# A Data-Centric Approach for Financial Large Language Models with Abductive Augmentation Reasoning

## Abstract

Large language models (LLMs) show promise for natural language tasks but struggle when applied directly to complex domains like finance. LLMs have difficulty reasoning about and integrating all relevant information. We propose a data-centric approach to enable LLMs to better handle financial tasks. Our key insight is that rather than overloading the LLM with everything at once, it is more effective to preprocess and pre-understand the data. We create a financial LLM (FLLM) using multitask prompt-based finetuning to achieve data pre-processing and pre-understanding. However, labeled data is scarce for each task. To overcome manual annotation costs, we employ abductive augmentation reasoning (AAR) to automatically generate training data by modifying the pseudo labels from FLLM's own outputs. Experiments show our data-centric FLLM with AAR substantially outperforms baseline financial LLMs designed for raw text, achieving state-of-the-art on financial analysis and interpretation tasks. We also open source a new benchmark for financial analysis and interpretation. Our methodology provides a promising path to unlock LLMs' potential for complex real-world domains.

## 1 Introduction

Large language models (LLMs) such as GPT-3 (Brown et al., 2020), GPT-4 (OpenAI, 2023), and Llama (Touvron et al., 2023) have revolutionized natural language processing tasks, excelling in text understanding, reasoning, and human-like response generation. While general LLMs are trained on broad corpora to acquire general knowledge about language, recent research (Li et al., 2023; Wu et al., 2023; Yang et al., 2023) has explored developing domain-specific LLMs by incorporating knowledge from specific fields. Domain-specific LLMs aim to achieve superior performance on domain-relevant tasks compared to general LLMs. Strategies like fine-tuning, prompt-based tuning, and in-context learning have been employed to incorporate domain knowledge into LLMs. The core challenge is developing effective techniques to inject the right domain knowledge into the LLMs and align their Language Modeling objective with domain-specific goals (Chu et al., 2023).

LLMs' attempt to directly access and utilize all domain knowledge in one shot is unrealistic. There are two main approaches to injecting knowledge into LLMs with or without additional training. One is to utilize prompt engineering to conduct in-context learning without any training, inserting information into prompts. However, token limitations arise when cramming excessive prompts into the context. Although tools like LangChain (Wu et al., 2022) can utilize embeddings instead of raw text, embedding provides a less direct means to integrate such external knowledge sources. They are limited in representing more complex conceptual relationships that are clear from linguistic context. A second technique involves leveraging new data to further train the large language model, fine-tuning its parameters on specific domains or tasks in order to adapt it for improved performance. While fine-tuning the large language model on new data can enhance its capabilities for certain applications, this approach has limitations in scale. As the model grows ever larger and more data is generated continuously, it becomes infeasible to retrain the model on all new information.

Therefore, in our work, we take the finance domain as an example. To enable language models to reason like financial experts, they must comprehend financial information multifariously. This necessitates integrating assorted tasks to acquire domain knowledge, such as event matching and analogy, assessing viewpoint quality, and extracting key points, among others. Thus, we propose a

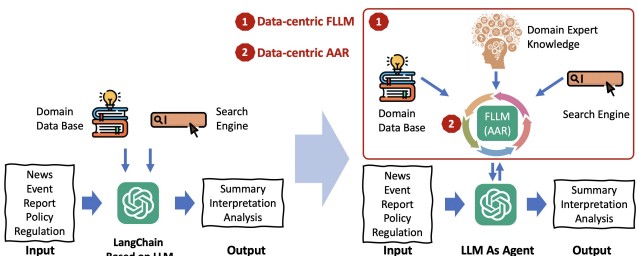

Figure 1: Our framework utilizes two key components - a large language model (FLLM) trained on financial data to preprocess domain-specific texts and an abductive reasoning module that augments data to improve FLLM. This differs from LangChain which operates directly on raw text corpora without any deep understanding and analysis of the raw financial data.

data-centric financial large language model named FLLM in Figure 1, based on a multitask prompt-based finetuning to achieve these different objectives. However, labeled data is limited for each specialized task in the complex financial domain, and annotators without domain expertise cannot competently label such data. We employ abductive learning to automatically generate training data by modifying pseudo labels from fledgling FLLM's own outputs to overcome the high cost of expert manual annotation. Our framework is highly adaptable, enabling the development of knowledgeable assistants across many domains. In summary, our proposed *data-centric* AI approach has two key facets. First, the financial knowledge base provides large language models with a preprocessed and parsed text corpus via data-centric FLLM. Second, abductive augmentation reasoning (AAR) addresses the scarcity of labeled data for specialized tasks to help train the FLLM. This combination of a financial large language model and abductive learning enables both knowledge injection into large language models and more sophisticated reasoning by conducting complex domain-specific tasks. The adaptable data-centric framework paves the way for knowledgeable AI assistants across finance and many other specialized fields.

## 2 BACKGROUND

### 2.1 IN-CONTEXT LEARNING

Large language models (LLMs) such as GPT-3 (Brown et al., 2020), GPT-4 (OpenAI, 2023), and Llama (Touvron et al., 2023) have shown impressive performance on a wide range of natural language tasks through a method known as in-context learning (Brown et al., 2020). This approach differs from traditional supervised learning which requires large labeled datasets. Instead, in-context learning allows models to acquire new skills and knowledge simply by being exposed to demonstrations of the task framed as natural language prompts (Liu et al., 2023). By conditioning the model on new prompts that provide examples, LLMs can exhibit zero-shot capabilities ranging from translation and summarization to mathematics and dialog, without updating the model parameters (Lu et al., 2021). Our work on abductive augmentation reasoning also relies on prompt-based in-context learning, with three core modules that leverage this technique to enable intuitive reasoning.

### 2.2 MULTITASK PROMPT-BASED FINETUNEING

By providing input-output examples as prompts, GPT-3 (Brown et al., 2020) showed an ability to solve NLP problems without full fine-tuning. This led to many prompt design techniques following a "pre-train, prompt, and predict" approach (Liu et al., 2021b). Some methods (Jiang et al., 2020; Shin et al., 2020; Liu et al., 2021a; Gao et al., 2021; Lu et al., 2022) search over discrete prompts, while others use continuous vector embeddings. Instruction-based prompts are more flexible and natural, containing detailed task descriptions. As human-like prompts enable learning from crowd-sourced data, instruction tuning of large language models is a promising approach for general NLP capabilities (Weller et al., 2020; Efrat & Levy, 2020). Similar to Geng et al. (2023), our work uses multi-task prompt finetuning on a financial corpus for data preprocessing and understanding, which unifies various financial subtasks in a shared model.

### 2.3 ABDUCTIVE REASONING

Reasoning is the process of using logic to draw conclusions based on available information (Wang et al., 2023). There are three main types of reasoning: deductive, inductive, and abductive. Deduc-

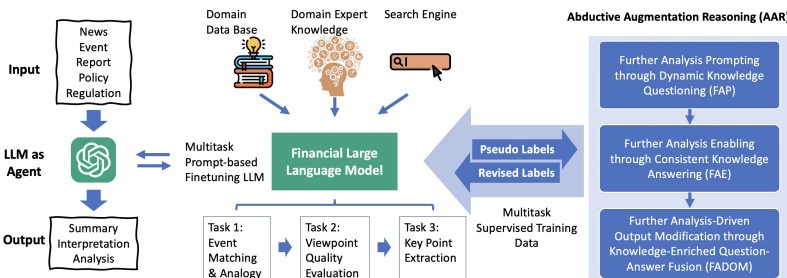

Figure 2: The framework of the financial large language model (FLLM), which specifically preprocesses the original corpus information, so as to establish a bridge between the input to be analyzed and the knowledge sources. Small labeled datasets are insufficient for finetuning large FLLM. AAR corrects pseudo labels from the fledgling FLLM to augment the labeled training data.

tive reasoning involves starting with a general premise or theory and drawing a specific conclusion based on that premise. Inductive reasoning works in the opposite direction - moving from specific observations to a general conclusion that is probable but not certain based on the evidence. Finally, abductive reasoning (Walton, 2014; Kovács & Spens, 2005; Zhou, 2019) starts with an observation and then seeks the simplest explanation that would lead to that observation. It generates hypotheses to explain a phenomenon rather than drawing conclusions. For example, upon observing that the grass is wet, one could abductively reason that it rained last night as a possible explanation. Abductive reasoning is useful for generating theories and new insights that can then be tested.

Our approach leverages the semantic reasoning abilities of large language models to augment training data through abductive inference. Rather than relying on symbolic rule formulations, we directly prompt the model with natural language descriptions of reasoning tasks. Recent work has shown that large language models learn rich semantic representations that allow them to make plausible inferences in context, despite lacking explicit symbolic reasoning capabilities (Tang et al., 2023). This pseudo-logical reasoning emerges from the models' ability to build robust connections between tokens, forming chains of reasoning that appear logically sound on the surface. Our method provides a more scalable approach to dataset augmentation through abductive logic compared to previous methods that require hand-crafted symbolic knowledge bases (Zhong et al., 2023).

## 3 METHODOLOGY

### 3.1 PROBLEM STATEMENT

Large language models (LLMs) have demonstrated impressive capabilities across a variety of domains, enabling applications for medical diagnosis and legal assistance. However, LLMs still struggle with complex reasoning and analysis tasks that require understanding, reasoning, and integrating information from diverse sources. This limitation is particularly evident in specialized domains like finance, where interpreting events, news, policies, and regulations requires integrating nuanced domain knowledge, synthesizing insights from multiple sources, elaborating logical reasoning, and generating an insightful point of view. In this work, our proposed system includes one fine-tuned financial large language model with access to external knowledge sources such as search engines, domain databases, and expert systems. This allows conducting financial sub-tasks to provide materials in a data-centric manner for final frozen LLM generation. Our ultimate goal is to utilize this deeply processed corpus to produce sophisticated financial analysis and interpretations. While we focus on financial analytics, our approach is designed to be generalizable across domains that require abundant information and complex reasoning.

### 3.2 DATA-CENTRIC FINANCIAL LARGE LANGUAGE MODEL

For the financial analysis and interpretation task, unlike the plain LangChain framework directly utilizing the raw information from different data sources, we establish one financial large language model (FLLM), which specifically preprocess the original corpus information, so as to establish a bridge between the input to be analyzed and the knowledge sources, including domain expert knowledge, financial databases, and search engines. As shown in Figure 2, our designed FLLM is a multitask prompt-based fine-tuning LLM that is designed to handle three key subtasks for this

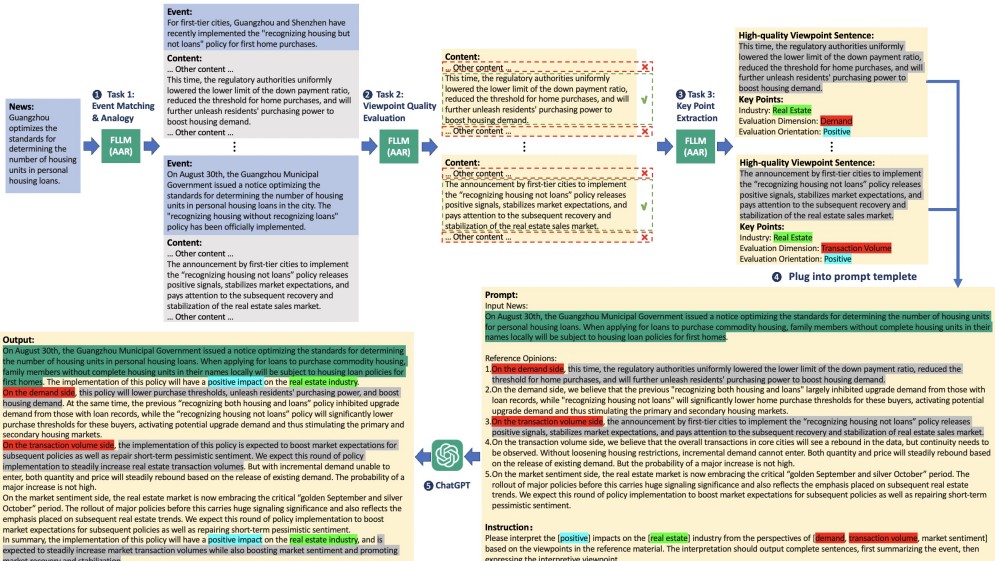

Figure 3: The example to instantiate the workflow of FLLM and the specific role of each subtask.

financial analysis and interpretation task, i.e., (1) event matching and analogy, (2) viewpoint quality evaluation, and (3) key point extraction. The model first matches the input news to relevant materials in the databases and finds analogous cases and reports. Then, the matched materials are evaluated for quality of opinion and analysis. Only the most insightful sentences are selected. Finally, the model extracts key details like industry, evaluation dimensions, sentiment, etc. to construct the structure for text generation. In this way, our financial large language model acts as an interpretive bridge between the input text and background knowledge sources. By preprocessing the data and learning correlations between events, viewpoints, and details, it can better leverage the information to produce high-quality financial analyses.

Specifically, we will use the example in Figure 3 to instantiate the end-to-end workflow of FLLM and the specific role of each subtask. The input is a new piece of government financial policy, about Guangzhou optimizes the standards for determining the number of housing units in personal housing loans. Firstly, we use a sub-ability of FLLM to match this financial policy with more materials, and get more analysis reports, although they may be inaccurate, scattered, or biased. Next, in step 2, FLLM selects the most insightful statements from this information and scores them to filter out irrelevant noise and distills the content down to concise prompts suitable for the language model's length limits later on. step 3, FLLM extracts high-level key information, such as industry, main indicators, analysis perspectives, sentiment, etc., to grasp and guide the direction, angle, and tone (positive or negative) for generating coherent text later. Through this series of FLLM modules, refined, focused, and filtered textual data has been prepared. In step 4, all this pre-processed information is formatted into a prompt template. Finally, in step 5, a large language model like ChatGPT utilizes this refined prompt to fluently generate useful interpretation and analysis of the policy's implications. By systematically preparing and guiding the input in a data-centric workflow, FLLM enables the final language model to produce focused, logical explanations of new financial policies. The end result is a cogent analysis specifically tailored to the original policy statement.

## 3.3 DATA-CENTRIC ABDUCTIVE AUGMENTATION REASONING

The workflow of the Financial Large Language Model has been detailed, but training such a multi-task prompt-based fine-tuning system poses challenges. These three financial subtasks demand strong domain knowledge, beyond what typical annotators possess. Thus, our labeled data is severely limited for these subtasks. Small labeled datasets are insufficient for finetuning large models. We must expand the data in a scalable way to improve the FLLM's performance. Although large language models show promise for text annotation (Dai et al., 2023), complex professional tasks remain difficult. Empirically, we have demonstrated that ChatGPT and GPT-4 struggle with these financial annotation tasks in the following experimental section. More advanced methods are needed to obtain quality labeled data. With better and more labeled data, the potential of FLLM can be realized for specialized subtasks.

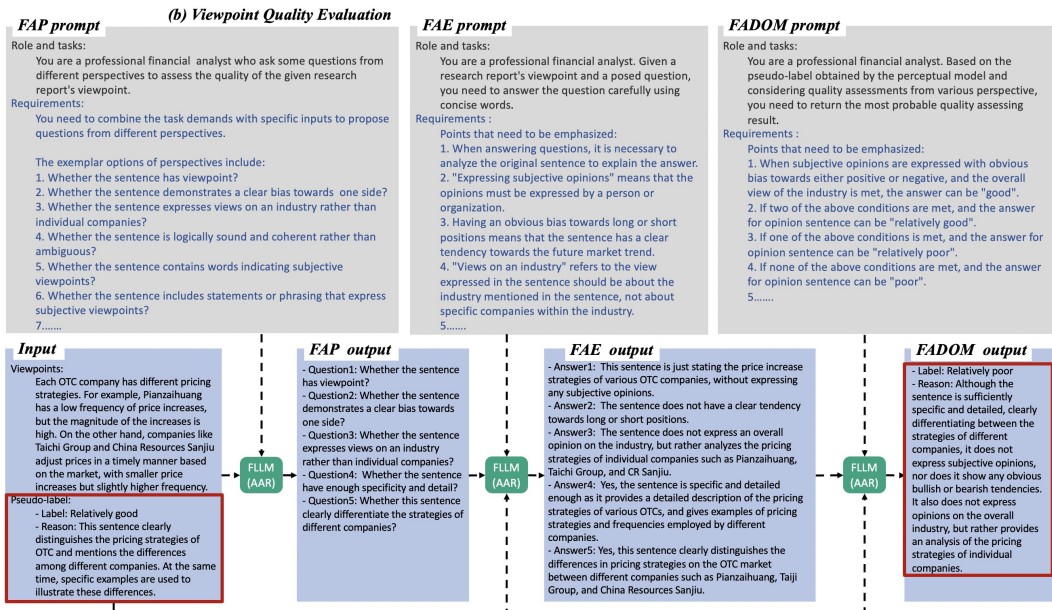

Figure 4: The example of AAR on viewpoint quality evaluation task. The examples of AAR on event matching and analogy and key point evaluation tasks are provided in the Appendix.

## 3.4 FRAMEWORK OF AAR

We propose an Abductive Augmentation Reasoning (AAR) algorithm to augment the training data for our fledgling FLLM in an abductive manner. The AAR takes as input the pseudo-labels produced for unlabeled data by the FLLM, which was trained on a small labeled dataset. These generated labels from fledgling FLLM may be erroneous due to the limited training data, making it challenging to achieve strong performance. To address this, the AAR refines the pseudo-labels through three key modules, i.e., Further Analysis Prompting through Dynamic Knowledge Questioning (FAP), Further Analysis Enabling through Consistent Knowledge Answering (FAE), and Further Analysis-Driven Output Modification through Knowledge-Enriched Question-Answer Fusion (FADOM). These three modules are driven by LLM such as ChatGPT or GPT-4 and interact with domain expert knowledge to refine the preliminary pseudo-labels, aiming to enhance the fledgling model's performance. This abductive reasoning process is used to correct faulty labels and provide higher-quality training data.

**FAP.** Further Analysis Prompting through Dynamic Knowledge Questioning (FAP) takes the original input text, the initial output predictions from the fledgling FLLM, and domain expert knowledge as inputs. FAP automatically generates a comprehensive series of analysis questions that aim to address any gaps, inconsistencies, or need for clarification in the fledgling FLLM's output. These questions are dynamically generated based on the specific output, prompting further reasoning and exploration. Example analysis questions can request more details on ambiguous conclusions, ask for the reasoning or evidence behind claims, probe hypothetical scenarios to check consistency, identify missing links in an argument, etc. The key is producing questions tailored to the output that can elicit a more complete, well-reasoned analysis when answered. Answering these questions will prompt further reasoning and lead to a more complete, logical analysis.

**FAE.** Further Analysis Enabling through Consistent Knowledge Answering (FAE) takes the original input text, the fledgling FLLM's initial output, the analysis questions from FAP, and the domain knowledge as inputs. FAE answers the analysis questions in a robust, consistent manner based on the domain knowledge. This provides broader, logically valid reasoning that aligns with known facts, relationships, and rules in the domain. FAE ensures the analysis is expanded in a knowledge-grounded way to fully address the gaps identified by the FAP questions.

**FADOM.** Further Analysis-Driven Output Modification through Knowledge-Enriched Question-Answer Fusion (FADOM) takes the original input, the fledgling FLLM's initial output, the analysis

questions and answers from FAP and FAE as inputs. FADOM selectively fuses the original output with the question-answer pairs in a way that incorporates the expanded analysis, reasoning, clarifications, and details provided by the QA process. This produces an improved output that benefits from abductive augmentation. The result is a more complete output aligned with domain expertise.

In summary, the automated AAR framework leverages abductive learning and dynamic QA over knowledge to augment FLLM's training data. This drives the fledgling FLLM to make more well-reasoned, detailed outputs consistent with the domain. As shown in Figure 4, the detailed prompt design, domain knowledge, input, and output of these three subtasks are provided, which shows that the three modules work together to enable systematic enhancement for each subtask.

## 4 EXPERIMENTS

In this section, we conduct experiments to evaluate the effectiveness of data-centric FLLM to enhance the generation by preprocessing the corpus information and data-centric AAR to improve FLLM by providing higher-quality and more training data. Specifically, we aim to address the following research questions:

1. Does AAR provide higher-quality data augmentation compared to annotations generated solely by large language models?

2. Can AAR boost performance on key financial subtasks addressed by our Financial Large Language Model?

3. Can providing pre-processed financial text data to LangChain through a financial language model lead to better financial analysis and interpretation compared to giving LangChain access to only raw financial text data?

Through these experiments, we aim to demonstrate that abductive reasoning based on LLM is an effective technique for data augmentation and model improvement. Further, the preprocessing and deep processing of corpus information in a data-centric manner is necessary and critical for complex text understanding, analysis, reasoning, and generation tasks in the field of expertise, such as finance.

### 4.1 DATASET AND TASK

The data were obtained from three main sources - web crawling (real-time storage of high-quality public opinion and analysis from across the web), purchasing (procurement of industry-specific analytical reports and exclusive information), and in-house data (large amounts of user discussions, influencer perspectives, and high-quality works accumulated within the platform ecosystem). Tens of millions of text corpus are stored daily. We also open source a new benchmark for financial analysis and interpretation. In this work, we take three financial subtasks as examples. **Event matching and analogy.** This task involves matching input news to relevant materials in databases to find analogous cases and reports. Evaluation metrics are precision, recall, and F1 score. These metrics measure the accuracy of matching input news to relevant materials. Higher scores indicate better performance. **Viewpoint quality evaluation.** This task evaluates the quality of opinion and analysis in the matched materials. Only the most insightful sentences are selected. The evaluation metric is classification accuracy. Measures how accurately the model classifies sentence quality into 2 or 4 categories like good/bad or excellent/good/fair/poor. Higher accuracy indicates better performance. **Key point extraction.** This task extracts key details like industry, evaluation dimensions, sentiment etc from materials to construct text summaries. Evaluation metrics are accuracy and BLEU score. Accuracy measures the correct extraction of key points. BLEU measures how close the constructed summary is to a human reference summary. Higher scores indicate better performance.

### 4.2 QUESTION 1: DOES AAR PROVIDE HIGHER-QUALITY DATA AUGMENTATION?

To answer whether abductive augmentation reasoning (AAR) provides higher-quality data augmentation compared to annotations generated solely by large language models, we designed a series of experiments to compare the annotation effects of AAR versus directly using existing large language models for annotation. We used ChatGPT and GPT-4 respectively to directly annotate 1000 unlabeled data points for each of three tasks: (1) event matching and analogy (EMA), (2) viewpoint quality evaluation (VQE), and (3) key point extraction (KPE).

Since our AAR includes three modules, and each module is built on top of the LLM, in order to explore the effects of different foundation models on AAR annotation, we also conducted a series of ablation studies, using ChatGPT, GPT-4, ChatGLM, ChatGLM2, Alpaca2, and LLama2 respectively as the foundation model for AAR. From Table 1, we can observe that simply using large language models makes it difficult to achieve annotation for these three complex financial tasks, while our AAR based on GPT-4 achieved the best results. In addition, we can see that AARs built on ChatGLM, ChatGLM2, Alpaca2, and LLama2 have difficulty directly running through the entire AAR workflow, with more or less issues existing, leading to the abductive reasoning process being unable to proceed smoothly. In summary, our experiments demonstrate that AAR can provide higher quality and more robust annotations compared to solely using LLMs, especially for complex domain-specific tasks. The choice of foundation model is also important, with more capable LLMs like GPT-4 better supporting the reasoning capabilities of AAR. There are still challenges in successfully implementing end-to-end abductive reasoning across different LLMs that require further research.

There are three modules in abductive augmentation reasoning (AAR), namely FAP, FAE, and FADOM. We incorporated domain expert knowledge to guide each of these three modules. To further explore the impact of AAR on data annotation and the role of domain expert knowledge in each module, we designed a series of experiments. As shown in Table 2, one or two modules contain expert knowledge to verify the impact of their knowledge on the overall AAR annotation results. From the table, we can observe that domain expert knowledge is useful for all three modules - removing any one of them affects the AAR annotation performance. The experiments provide insights into how expert knowledge can be effectively incorporated into AAR to improve its data annotation capabilities. This allows AAR to be customized for different domains by plugging in relevant knowledge bases. Overall, explicitly encoding domain knowledge is shown to be an important aspect of developing robust AAR systems.

Table 1: The comparison of AAR data augmentation and direct annotation by LLM.

| Settings | | | KPE | | | VQE | | EMA | |
| --- | --- | --- | --- | --- | --- | --- | --- | --- | --- |
| Strategy | Base Model | Prompt | Precision | Recall | F1 | Accuracy(2) | Accuracy(4) | Accuracy | BLEU |
| Direct annotation | ChatGPT | 1 shot | 0.014 | 0.023 | 0.018 | 0.47 | 0.21 | 0.67 | 0.399 |
| | GPT-4 | 1 shot | 0.009 | 0.016 | 0.011 | 0.60 | 0.22 | 0.80 | 0.482 |
| AAR | ChatGPT | 1 shot | 0.004 | 0.008 | 0.005 | 0.52 | 0.32 | 0.75 | 0.316 |
| | GPT-4 | 1 shot | **0.226** | **0.414** | **0.293** | **0.71** | **0.40** | **0.87** | **0.533** |
| | ChatGLM | 1 shot | - | - | - | - | - | - | - |
| | ChatGLM2 | 1 shot | - | - | - | - | - | - | - |
| | Alpaca2 | 1 shot | - | - | - | - | - | - | - |
| | LLama2 | 1 shot | - | - | - | - | - | - | - |

Table 2: The influence of domain expert knowledge of three modules on the AAR performance.

| Settings | | KPE | | | VQE | | EMA | |
| --- | --- | --- | --- | --- | --- | --- | --- | --- |
| AAR | Knowledge | Precision | Recall | F1 | Accuracy(2) | Accuracy(4) | Accuracy | BLEU |
| GPT-4 | No | 0.000 | 0.000 | 0.000 | 0.40 | 0.14 | 0.78 | 0.465 |
| GPT-4 | FAP | 0.005 | 0.008 | 0.006 | 0.40 | 0.18 | 0.82 | 0.477 |
| GPT-4 | FAE | 0.041 | 0.062 | 0.050 | 0.42 | 0.15 | 0.84 | 0.496 |
| GPT-4 | FADOM | 0.042 | 0.070 | 0.053 | 0.58 | 0.27 | 0.82 | 0.504 |
| GPT-4 | FAP+FAE | 0.027 | 0.039 | 0.032 | 0.36 | 0.15 | 0.87 | 0.511 |
| GPT-4 | FAP+FADOM | 0.029 | 0.047 | 0.036 | 0.56 | 0.33 | 0.84 | 0.483 |
| GPT-4 | FAE+FADOM | 0.163 | 0.234 | 0.192 | 0.59 | 0.36 | 0.84 | 0.520 |
| GPT-4 | All | **0.226** | **0.414** | **0.293** | **0.71** | **0.40** | **0.87** | **0.533** |

### 4.3 QUESTION 2: CAN AAR BOOST THE PERFORMANCE OF OUR FLLM?

To explore whether AAR can boost performance on key financial subtasks addressed by our Financial Large Language Model, we designed three strategies with our FLLM. First, we leveraged state-of-the-art general-purpose large language models like ChatGPT and GPT-4 without any training, using prompt engineering with one-shot and few-shot demonstrations to guide the FLLM on the three financial tasks. Second, we fine-tuned the openly available large language models on a small amount of expert-annotated financial data. Third, we utilized the AAR technique to augment the small amount of expert-labeled data into a larger high-quality labeled dataset for fine-tuning our FLLM foundation model.

Table 3: The performance comparison of different training strategies of FLLM on three tasks. **Red**: the best, Blue: the second best.

| Settings | | | KPE | | | VQE | | EMA | |
|---|---|---|---|---|---|---|---|---|---|
| Strategy | FLLM | Prompt | Precision | Recall | F1 | Accuracy(2) | Accuracy(4) | Accuracy | BLEU |
| No training | ChatGPT | 1 shot | 0.014 | 0.023 | 0.018 | 0.47 | 0.21 | 0.67 | 0.399 |
| | GPT-4 | 1 shot | 0.009 | 0.016 | 0.011 | 0.60 | 0.22 | 0.80 | 0.482 |
| | ChatGPT | 20 shots | 0.179 | 0.203 | 0.190 | 0.52 | 0.32 | 0.75 | 0.357 |
| | GPT-4 | 20 shots | 0.245 | 0.266 | 0.255 | **0.71** | **0.49** | **0.84** | **0.535** |
| Finetune | ChatGLM | 1 shot | 0.057 | 0.047 | 0.052 | 0.53 | 0.30 | 0.60 | 0.328 |
| | ChatGLM2 | 1 shot | 0.093 | 0.133 | 0.109 | 0.50 | 0.36 | 0.60 | 0.353 |
| | Alpaca2 | 1 shot | 0.160 | 0.164 | 0.162 | 0.57 | 0.34 | 0.55 | 0.295 |
| AAR + Finetune | ChatGLM | 1 shot | **0.260** | 0.305 | **0.281** | 0.63 | 0.26 | 0.68 | 0.379 |
| | ChatGLM2 | 1 shot | 0.182 | 0.344 | 0.238 | 0.62 | 0.34 | 0.67 | 0.389 |
| | Alpaca2 | 1 shot | 0.209 | **0.367** | 0.266 | 0.69 | 0.39 | 0.83 | 0.485 |

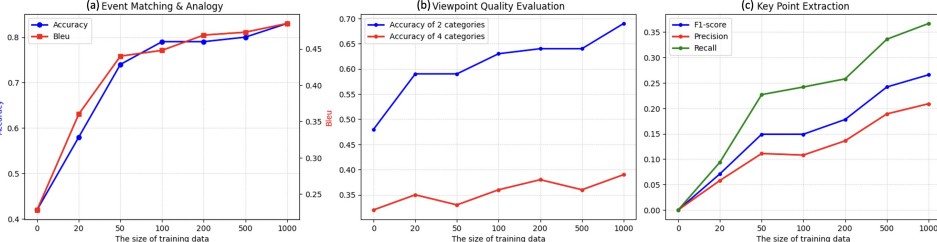

Figure 5: The performance of fine-tuned FLLMs as the amount of labeled training data increases.

As shown in Table 3, While GPT-4 with 20 shots prompting demonstrates impressive capabilities out-of-the-box, our approach of applying AAR data augmentation and then fine-tuning tailors the model more specifically to the financial domain. This allows our FLLM to reach comparable performance to GPT-4 on the key metrics across all three financial analysis subtasks. The augmented training dataset created through AAR provides the FLLM with sufficient coverage of the problem space to match the few-shot generalization abilities of a cutting-edge general-purpose LLM like GPT-4. Our results highlight the potential of targeted data augmentation techniques like AAR to unlock specialized performance in limited resource contexts where acquiring substantial direct human annotations is infeasible. With further development, AAR data augmentation could enable high-performance financial LLMs without the need for massive human labeling efforts. The key advantage of AAR is that it provides an efficient way to generate more labeled data from a small seed set, which is especially valuable in specialized domains like finance where expert-labeled data is scarce. By leveraging AAR to amplify the limited human annotations, we were able to significantly boost our FLLM's performance on core financial analysis subtasks relevant to real-world applications.

Furthermore, to further explore the effects of abductive augmentation reasoning (AAR) on financial large language models (FLLMs), we conducted a series of experiments by annotating different amounts of FLLM training data with AAR annotations. We then fine-tuned the FLLMs and observed how their performance changed across all tasks and metrics as the amount of annotated data increased. The results, shown in Figure 5, demonstrate that metrics across all three tasks improved as more annotated data was used. This suggests that incorporating AAR into the training process can enhance FLLMs' reasoning and generalization abilities for financial applications. Specifically, AAR's iterative generation and evaluation of hypotheses appears to provide a form of inductive bias that helps the model better capture financial reasoning patterns and semantics from limited data. Overall, our experiments reveal the promise of AAR for imbuing FLLMs with more robust financial intelligence. Further research is warranted to determine optimal AAR annotation strategies and model architectures to maximize the financial reasoning capacity of large language models.

### 4.4 QUESTION 3: CAN FLLM HELP LANGCHAIN TO GENERATE BETTER OUTPUT?

We will evaluate LangChain's ability to provide insightful financial analysis and interpretations when given pre-processed via FLLM vs. raw financial text data, rating it on four dimensions: **Relevance (0-5)**: The analysis should focus on interpreting the core events described, without straying into unrelated topics or generic background. **Accuracy (0-5)**: The analysis's viewpoint and reasoning should seem reasonable. It should consistently express a positive or negative outlook, without exaggerating or downplaying the event's impact on the industry. **Logic (0-5)**: The analysis should

Table 4: The comparison of LangChain and our pipeline on financial analysis and interpretations.

| Metric | LangChain | FLLM w/ 1,2,3 | FLLM w/ 1 | FLLM w/ 1,2 |
|---|---|---|---|---|
| Relevance | $4.28 \pm 0.57$ | $\mathbf{4.85 \pm 0.14}$ | $4.42 \pm 0.61$ | $4.57 \pm 0.61$ |
| Accuracy | $4.14 \pm 1.14$ | $\mathbf{4.78 \pm 0.15}$ | $4.35 \pm 0.55$ | $4.50 \pm 0.25$ |
| Logic | $3.71 \pm 0.23$ | $\mathbf{4.28 \pm 0.23}$ | $3.42 \pm 0.28$ | $3.57 \pm 0.62$ |
| Expertise | $3.57 \pm 0.28$ | $\mathbf{4.71 \pm 0.23}$ | $3.78 \pm 0.15$ | $3.85 \pm 0.14$ |

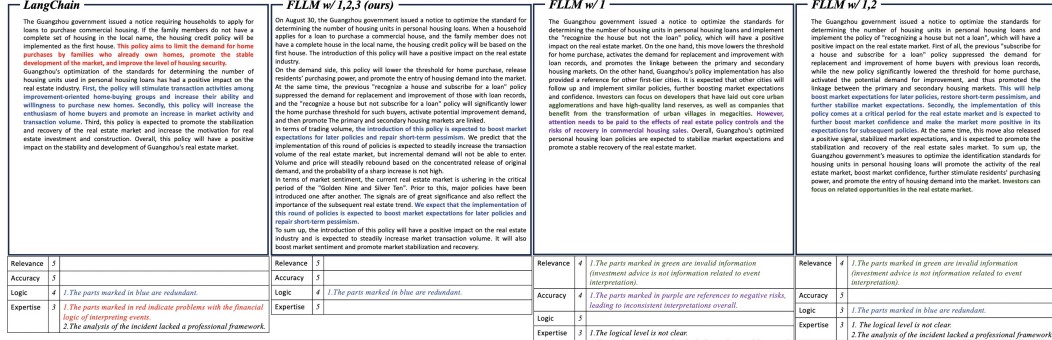

Figure 6: Real financial analysis and interpretation examples with detailed reasons and scores.

flow smoothly and logically, with clear causality and relationships between points. It should not simply restate event details or repeat the same point in different words. The overall meaning should be coherent and well-structured. **Expertise (0-5)**: The analysis should examine the event's impacts from multiple professional investing angles. It should demonstrate sound financial logic and insightful consideration of how the event could affect valuations. There should be a clear, layered structure to the interpretation.

To robustly evaluate the capabilities of plain LangChain versus enhanced LangChain via FLLM, we conducted a rigorous comparative experiment. 1000 recent news articles were analyzed and interpreted using both plain LangChain and LangChain enhanced with the FLLM. To obtain objective assessments, five independent human annotators were then invited to carefully review the 1000 sample outputs across the four dimensions mentioned above. By averaging the annotators' scores in each dimension, we could quantify the improvements afforded by integrating FLLM into LangChain in an unbiased, statistically-sound manner. From Table 4, we observed that our method significantly outperformed plain LangChain on all metrics.

Additionally, to evaluate the contribution of our 3 subtasks of FLLM - (1) event matching and analogy, (2) viewpoint quality evaluation, and (3) key point extraction - we designed 2 additional ablation studies. In our original design (FLLM w/ 1,2,3), the outputs from all 3 subtasks are injected into the final prompt of ChatGPT to guide generation. In the first ablation study (FLLM w/ 1), we only input the results from subtask 1 on event matching and analogy, containing only the matched corpus resources. In the second ablation study (FLLM w/ 1,2), we input the results from subtask 1 and 2, including the matched corpus resources and high-quality viewpoints selected. From the results, we observed that all 3 subtasks play necessary and complementary roles in producing the final generated text. In addition, as shown in Figure 6, we give a real example with detailed reasons.

## 5 CONCLUSION AND FUTURE WORK

This paper proposes a data-centric approach based on FLLM to improve LLMs' capabilities on financial analysis tasks. To overcome the scarcity of labeled data, they employ abductive augmentation reasoning to automatically generate training data. Experiments demonstrate their data-centric financial LLM with abductive augmentation reasoning substantially outperforms baseline LLMs, achieving state-of-the-art on financial analysis and interpretation benchmarks. The data-centric methodology provides a promising direction to unlock the potential of LLMs for complex real-world domains. The introduction of a new benchmark for financial analysis and interpretation is also a valuable contribution. Besides, an interesting direction for future work is to combine the data-centric approach with other methods like prompting and self-supervised pretraining on financial texts. Integrating multi-modal data like financial reports, earnings calls, and stock prices could also enable more nuanced financial analysis.

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
