# OpenReview forum: "A Data-Centric Approach for Financial Large Language Models with Abductive Augmentation Reasoning"
_ICLR.cc/2024/Conference — ICLR 2024 Conference Withdrawn Submission_

### Official Review · Reviewer_xmH1 · 2023-10-29

**Soundness:** 2 fair
**Presentation:** 2 fair
**Contribution:** 2 fair
**Rating:** 3
**Confidence:** 4

**Summary:**

This paper introduces a data-centric methodology, utilizing abductive data augmentation, to develop robust large language models tailored for financial applications. It addresses the pivotal issue of general large language models, like ChatGPT and GPT-4, generating imprecise labels when annotating financial texts. The authors incorporate domain-specific knowledge into three  abductive and pipelined reasoning modules, aiming to enhance the accuracy of pseudo labels. Empirical experiments, conducted in both no-training and fine-tuning scenarios,  show that the promising potential in certain specific direction of the proposed method (ARR).

**Strengths:**

1.  The use of abductive reasoning to augment small-scale training data is novel to me.
2.  The three abductive reasoning module includes question generation,  question answering and final answer generation is intuitive.

**Weaknesses:**

1. The reproducibility is a big issue for this paper.  First, the code is not released.  Second, the evaluation is only done on in-house datasets.  Third, the authors claims to open an evaluation benchmark, but there is no introduction about the released benchmark in this paper.
2. The writing is not very clear.  This paper uses “domain expert knowledge” in multiple locations, without explaining what kind of knowledges they are and in what kind of format is unknown.  Do you mean the prompts are the domain expert knowledge?
3. Combining the results in Table 1 and Table 3,  ARR (GPT-4) + Finetune  does not show advantages than GPT4 (20 shots), especially on VQE and EMA datasets,  which largely weakens the claim and contribution of this paper.  Based on these observations, why do not we directly use GPT4 (20 shots) to annotate the datasets, instead of using manually designed prompts?
4. Table 1 only show 1 shot results.  More results are necessary.
5. Some settings of the fine-tuning are not clear.  See questions.
6. There are no comparisons to open-sourced financial large language models such as FinGPT and FinMA.

**Questions:**

1. For the fine-tuning settings, how many examples are annotated?
2. Why do you choose these three tasks? Are they specifically fitted for this research question?
3. For the open-sourced benchmark for financial analysis and interpretation,  more details are necessary.
4. In Table 3,  what are the performances when you do ARR with GPT4 (20-shots) first and then finetune?

---

### Official Review · Reviewer_2L6F · 2023-10-30

**Soundness:** 3 good
**Presentation:** 3 good
**Contribution:** 2 fair
**Rating:** 3
**Confidence:** 4

**Summary:**

The paper focuses on developing high-quality financial large language models (FLLMs). The authors propose a digest-then-process framework that first utilizes FLLMs to digest data and generate informative prompts by event matching & analogy, viewpoint quality evaluation, and key point extraction together with external expert domain knowledge. Then, the informative prompts are sent to LLMs to generate final answers. To train FLLMs and overcome the data scarcity problem, authors use abductive augmentation reasoning (AAR) to augment data, which consists of three-step prompting: Further Analysis Prompting through Dynamic Knowledge Questioning (FAP), Further Analysis Enabling through Consistent Knowledge Answering (FAE) and Further Analysis-Driven Output Modification through Knowledge-Enriched Question-Answer Fusion (FADOM). The authors conduct experiments to show the effectiveness of AAR and the inference pipelines.

**Strengths:**

1. The proposed inference pipeline and AAR are effective based on the experiments.
2. The methods are well-illustrated and easy for readers to understand. I can understand this paper quite well even I am not an expert in finance.

**Weaknesses:**

1. I suspect that the current approach can train better domain-specific models. The critical component of AAR is prompting LLMs, which limits the upper bounds of trained FLLMs. That is to say, FLLMs may not be able to outperform LLMs since LLMs provide labels, and I doubt the performance will be higher if we change FLLMs to LLMs. Nevertheless, I acknowledge the prompting techniques for both inferences and AAR are effective, and I think the major advantage of the current approach is effectively distilling domain-specific knowledge from LLMs, rather than training domain-specific models that outperform LLMs.
2. The current inference contains three stages: event matching & analogy, viewpoint quality evaluation, and key point extraction. I wonder if they are sufficient for the whole (or the majority of) the finance domain, or just a specific task? If it is the latter case, I think the application is limited and would like to know if there are methods to extend the current approach to other tasks.
3. The paper lacks many details, such as FLLM model sizes, training data sizes, and other necessary training/evaluation details for replication. I would suggest authors to add these details if there are no commercial concerns.

**Questions:**

Please see the Weaknesses part.

**Details Of Ethics Concerns:**

No ethics concern for this paper.

---

### Official Review · Reviewer_e2qX · 2023-10-31

**Soundness:** 3 good
**Presentation:** 2 fair
**Contribution:** 3 good
**Rating:** 5
**Confidence:** 4

**Summary:**

This paper proposes a data-centric financial large language model (FLLM) based on a multitask prompt-based fine-tuning approach for the financial analysis and interpretation task. For the multitask prompt-based framework, the authors propose three key sub-tasks: event matching and analogy, viewpoint quality evaluation, and key point extraction. Through sequential prompting of these tasks, FFLM can learn better correlations between events, viewpoints, and details so as to generate higher-quality financial analysis.

Since the proposed three sub-tasks require strong domain knowledge, the authors propose an Abductive Augmentation Reasoning (AAR) algorithm for training data augmentation. AAR contains three stages of analytical questions: FAP, FAE, and FADOM.

The authors conduct experiments and ablation studies to demonstrate the effectiveness of the proposed modules.

**Strengths:**

1. The paper addresses an important research direction: how to build LLMs in specific domains, especially domains requiring high expertise and lack training data. The proposed FLLM can serve as a useful application for financial analysis.
2. The experiments show the proposed workflow and data augmentation technique is effective.

**Weaknesses:**

1. The proposed framework seems a bit ad-hoc for a specific type of financial document - the government financial policy used in this work. It is overclaiming to say "financial LLM". There are many tasks in financial NLP, e.g., question answering [1,2].
2. Lack of related work - A detailed discussion for finance NLP is necessary.
3. For the AAR part, since the FAP, FAE, and FADOM stages are all about raising questions, what if we just simply merge all the questions together instead of splitting into three stages?
4. The writing and clarity of the paper can be improved: The details of the dataset are not clear. What are the data sources? What's the size of the test data for each module? How much annotated data do you use for fine-tuning? How much data do you augment?

[1] TAT-QA: A Question Answering Benchmark on a Hybrid of Tabular and Textual Content in Finance, Fengbin Zhu, Wenqiang Lei, Youcheng Huang, Chao Wang, Shuo Zhang, Jiancheng Lv, Fuli Feng, Tat-Seng Chua, ACL 2021

[2] FinQA: A Dataset of Numerical Reasoning over Financial Data, Zhiyu Chen, Wenhu Chen, Charese Smiley, Sameena Shah, Iana Borova, Dylan Langdon, Reema Moussa, Matt Beane, Ting-Hao Huang, Bryan Routledge, William Yang Wang, EMNLP 2021

**Questions:**

For evaluating the final generated output, how do you hire human annotators? Do you use financial experts for the evaluation?

---

### Official Review · Reviewer_z1t5 · 2023-10-31

**Soundness:** 2 fair
**Presentation:** 2 fair
**Contribution:** 1 poor
**Rating:** 3
**Confidence:** 4

**Summary:**

This study introduces an abductive reasoning-driven data augmentation approach by collecting and modifying pseudo labels form a LLM (large language model) to generate training data. This strategy is employed to generate augmented training datasets, facilitating the Language Model's enhanced utilization of domain-specific knowledge for improved performance in specialized tasks. Abductive reasoning, known for providing a simple explanation for a given observation. The empirical investigations undertaken within the finance domain yield noteworthy outcomes, demonstrating that the trained financial LLM exhibits superior performance compared to various baseline models, with the exception of GPT-4 on some tasks.

**Strengths:**

(1) The experimental findings presented in this study demonstrate that the Language Model (LLM) trained using the proposed method attains a level of performance that approaches state-of-the-art across diverse financial datasets.

(2) The paper is easy to follow due to the authors' effective provision of requisite background information on abductive reasoning.

**Weaknesses:**

(1) The paper's novelty appears somewhat limited, as the utilization of abductive reasoning or similar techniques for data augmentation has already been extensively explored and established as standard practice across a multitude of tasks.

(2) The experimental results presented in Table 3 are insufficient in scope. A more comprehensive comparison with alternative data augmentation methods is needed to substantiate the efficacy of abductive augmentation reasoning (ARR) fully.

(3) Many typographical and grammatical errors are present in the manuscript, which detracts from its overall quality and readability.

**Questions:**

(1) What did you mean by “embedding provides a less direct means to integrate such external knowledge sources.”?

(2) The authors claim that their framework is highly versatile, facilitating the creation of knowledgeable LLMs in various domains. It would be valuable to investigate the performance of the proposed framework in domains beyond finance.

(3) What is the specific significance of abductive reasoning for tasks within the finance domain? An explanation of the rationale behind the choice of abductive reasoning as a crucial element in this context would enhance the paper's clarity.

(4) Could you provide insights into the selection of LangChain as the baseline model?